



# Technical Note: On the confounding similarity of two water
# balance formulas – Turc-Mezentsev *vs* Tixeront-Fu
Vazken Andréassian[1] & Tewfik Sari[2]
[(1)] Irstea, HYCAR Research Unit, Antony, France
[(2)] ITAP, Univ Montpellier, Irstea, Montpellier SupAgro, Montpellier, France
## Abstract
This Technical Note documents and analyzes the confounding similarity of two widely used
water balance formulas: Turc-Mezentsev and Tixeront-Fu. It details their history, their
hydrological and mathematical properties, and discusses the mathematical reasoning behind
their slight differences. Apart from the difference identified in their partial differential
expressions, both formulas share the same hydrological properties and it seems impossible
to recommend one over the other as more "hydrologically founded": hydrologists should feel
free to choose the one they feel more comfortable with.
## Keywords
Water balance formulas, Turc-Mezentsev formula, Tixeront-Fu formula, Budyko hypothesis
## 1.    Introduction
The Turc-Mezentsev (Mezentsev, 1955;Turc, 1954) and Tixeront-Fu (Fu, 1981;Tixeront,
1964) formulas were introduced to model long-term water balance at the catchment scale.
Both formulas are almost equivalent numerically (but differ nonetheless). Surprisingly,
comparisons are rare: Tixeront knew Turc (1954) work, which he cites, but it seems that he
did not realize that Turc's formulation was numerically equivalent to the one he proposed.
Similarly, Fu knew Mezentsev (1955) work because he starts his 1981 paper discussing it,
but it seems that he did not realize that the formulation he obtained was so close numerically.
As far as we know, Yang et al. (2008) were the first to compare the Turc-Mezentsev and the
Tixeront-Fu formulas and to conclude that both formulas were "approximately equivalent." In
this note we further elaborate the confounding similarity between the two formulas and
contribute complementary explanations on their underlying hypotheses.





## 2.    Presentation of the Turc-Mezentsev (TM) and the Tixeront-Fu (TF) formulas

The TM and TF formulas use as inputs long-term average precipitation $P$ [mm/yr] and long-term average maximum evaporation $E_0$ [mm/yr]. They produce as outputs either long-term average specific discharge $Q$ [mm/yr] or long-term average actual evaporation $E$ [mm/yr]. There are two formulations (one giving $Q$ as a function of $P$ and $E_0$ and one giving $E$ as a function of the same variables), equivalent under the assumption that the catchment is conservative (i.e., that it does not "leak" towards deep aquifers) so that $E$ and $Q$ can be linked through the equation $E = P - Q$. Maximum evaporation is understood in the sense of Budyko (1963 /1948/) as the water equivalent of the energy available to evaporation. In what follows, the $E_0/P$ ratio is called the aridity ratio, its inverse (i.e., the $P/E_0$ ratio) is called the humidity ratio. The formulas are presented in Table 1. Because none of the original papers introducing them are in English, we also briefly document their origins in the appendix.

**Table 1. Turc-Mezentsev (TM) and Tixeront-Fu (TF) water–energy balance formulations ($P$ – precipitation, $Q$ – streamflow, $E_0$ – maximum evaporation, $E$ – actual evaporation, all in mm/year averaged over many years). $n$ is the free parameter of the Turc-Mezentsev formula [$n > 0$]; $m$ is the free parameter of the Tixeront-Fu formula [$m > 1$].**

| Reference | Streamflow formulation | Actual evaporation formulation | Parameter |
|---|---|---|---|
| Turc (1954), Mezentsev (1955) | $Q = P - [P^{-n} + E_0^{-n}]^{\frac{-1}{n}}$ <br><br> Eq. 1 | $E = [P^{-n} + E_0^{-n}]^{\frac{-1}{n}}$ <br><br> Eq. 2 | $n > 0$ |
| Tixeront (1964), Fu (1981) | $Q = [P^m + E_0^m]^{\frac{1}{m}} - E_0$ <br><br> Eq. 3 | $E = P + E_0 - [P^m + E_0^m]^{\frac{1}{m}}$ <br><br> Eq. 4 | $m > 1$ |

We need to clarify here that the TM and TF formulas can be found in the hydrologic literature under different names. The naming convention we adopted is explained as follows: Eq. 1 and Eq. 2 are named "Turc-Mezentsev" (TM) because Turc (1954) and Mezentsev (1955) worked independently and published the same equation almost simultaneously. Eq. 3 and Eq. 4 are named "Tixeront-Fu" (TF) because although Tixeront's original publication predates Fu's by almost 20 years, both publications were independent, and the name of Fu has already gained wide international recognition. Both formulas are sometimes referred to as "Budyko-type," although none of them were actually used by Budyko (1963 /1948/), who instead used





a parameter-free formula derived from the work of Oldekop (1911) (for a synthesis of
Oldekop's work and how it was used by Budyko, see Andréassian et al., 2016). Other
authors have published papers containing the TM formula: see e.g. Hsuen-Chun (1988) and
Choudhury (1999), and their names are sometimes used to designate it.

In our interpretation of the TM and TF formulas, we will use their partial derivatives, which we
present in Table 2 and Table 3.

**Table 2. Partial derivatives of the Turc-Mezentsev formula ($P$ – precipitation, $Q$ – streamflow, $E_0$**
**– maximum evaporation, $E$ – actual evaporation, all in mm/year averaged over many years). $n$ is**
**the free parameter of the Turc-Mezentsev formula [$n > 0$]**

| Streamflow formulation | | Actual evaporation formulation | |
|---|---|---|---|
| $\dfrac{\partial Q}{\partial P} = 1 - \left(1 + \left(\dfrac{P}{E_0}\right)^n\right)^{-\frac{1}{n}-1}$ | **Eq. 5** | $\dfrac{\partial E}{\partial P} = \left(1 + \left(\dfrac{P}{E_0}\right)^n\right)^{-\frac{1}{n}-1}$ | **Eq. 6** |
| $\dfrac{\partial Q}{\partial E_0} = -\left(1 + \left(\dfrac{E_0}{P}\right)^n\right)^{-\frac{1}{n}-1}$ | **Eq. 7** | $\dfrac{\partial E}{\partial E_0} = \left(1 + \left(\dfrac{E_0}{P}\right)^n\right)^{-\frac{1}{n}-1}$ | **Eq. 8** |


**Table 3. Partial derivatives of the Tixeront-Fu formula ($P$ – precipitation, $Q$ – streamflow, $E_0$ –**
**maximum evaporation, $E$ – actual evaporation, all in mm/year averaged over many years). $m$ is**
**the free parameter of the Tixeront-Fu formula [$m > 1$]**

| Streamflow formulation | | Actual evaporation formulation | |
|---|---|---|---|
| $\dfrac{\partial Q}{\partial P} = \left(1 + \left(\dfrac{E_0}{P}\right)^m\right)^{\frac{1}{m}-1}$ | **Eq. 9** | $\dfrac{\partial E}{\partial P} = 1 - \left(1 + \left(\dfrac{E_0}{P}\right)^m\right)^{\frac{1}{m}-1}$ | **Eq. 10** |
| $\dfrac{\partial Q}{\partial E_0} = -1 + \left(1 + \left(\dfrac{P}{E_0}\right)^m\right)^{\frac{1}{m}-1}$ | **Eq. 11** | $\dfrac{\partial E}{\partial E_0} = 1 - \left(1 + \left(\dfrac{P}{E_0}\right)^m\right)^{\frac{1}{m}-1}$ | **Eq. 12** |


## 77 3. Comparisons of the Turc-Mezentsev and Tixeront-Fu formulas

### 78 3.1 Previous comparisons

We mentioned in the introduction that the first paper comparing the TM and TF formulas was
published by Yang et al. (2008), who note that the TM and TF formulas are "approximately
equivalent" and that their parameters have a "perfectly significant linear correlation
relationship," which they identify as in Eq. 13:




$$m \sim n + 0.72 \qquad \qquad \textbf{Eq. 13}$$

where $m$ stands for the parameter of the Tixeront-Fu relationship and $n$ for the parameter of
the Turc-Mezentsev relationship.
Note that Eq. 13 is an experimental relationship obtained by regression. It gives slightly more
satisfying results that the "theoretical" relationship (found by posing $P/E_0$=1 in both TM and
TF) given below (Eq. 14):

$$m = \frac{ln2}{ln\left[2 - 2^{\frac{-1}{n}}\right]} \qquad \qquad \textbf{Eq. 14}$$


Recently, Andréassian et al. (2016) and de Lavenne and Andréassian (2018) used the Yang
et al. (2008) results and further illustrated the nearly perfect similarity between the two
formulas.

**3.2    Graphical illustration of the similarity of the TM and the TF formulas**
Figure 1, which illustrates the confounding numerical proximity of the two formulas, speaks
for itself: while we tested a wide range of ($n,m$) couples respecting Eq. 13, the difference
(TM-TF) between the two formulas is at maximum 2.5%, and most of the time much less.
Numerically, both formulas are equivalent (except for very low values of the humidity index
$P/E_0$).





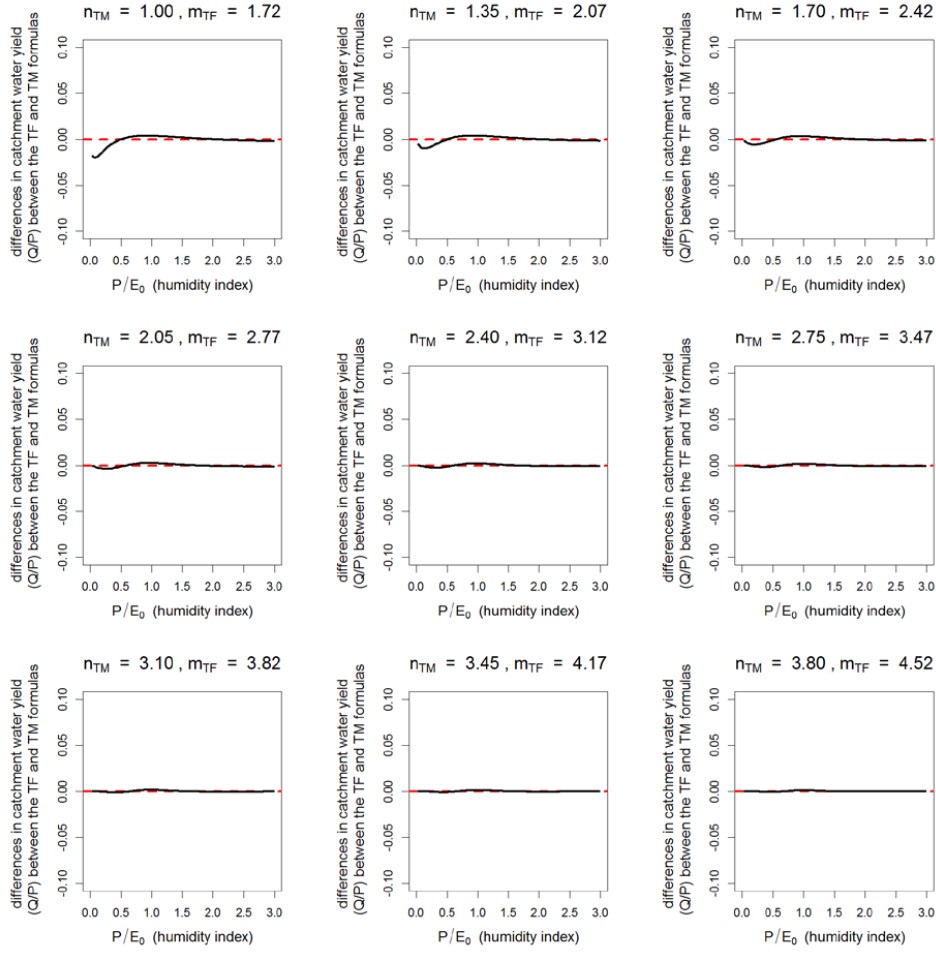


**Figure 1. Illustration of the similarity between the values of the Turc-Mezentsev (TM) and the**
**Tixeront-Fu (TF) formulas for a range of values of *n* (the parameter of the TM formula) and *m***
**(the parameter of the TF formula), using the Yang et al. (2008) relationship: *m = n + 0.72***


Figure 2 and Figure 3 also present the differences between the partial derivatives of the TM
and TF formulas. The reason for this is that both formulas are sometimes used to predict the
hydrological impact of climatic change, i.e., to evaluate the evolution or differences between
future and current conditions. Again, both formulas appear numerically equivalent.





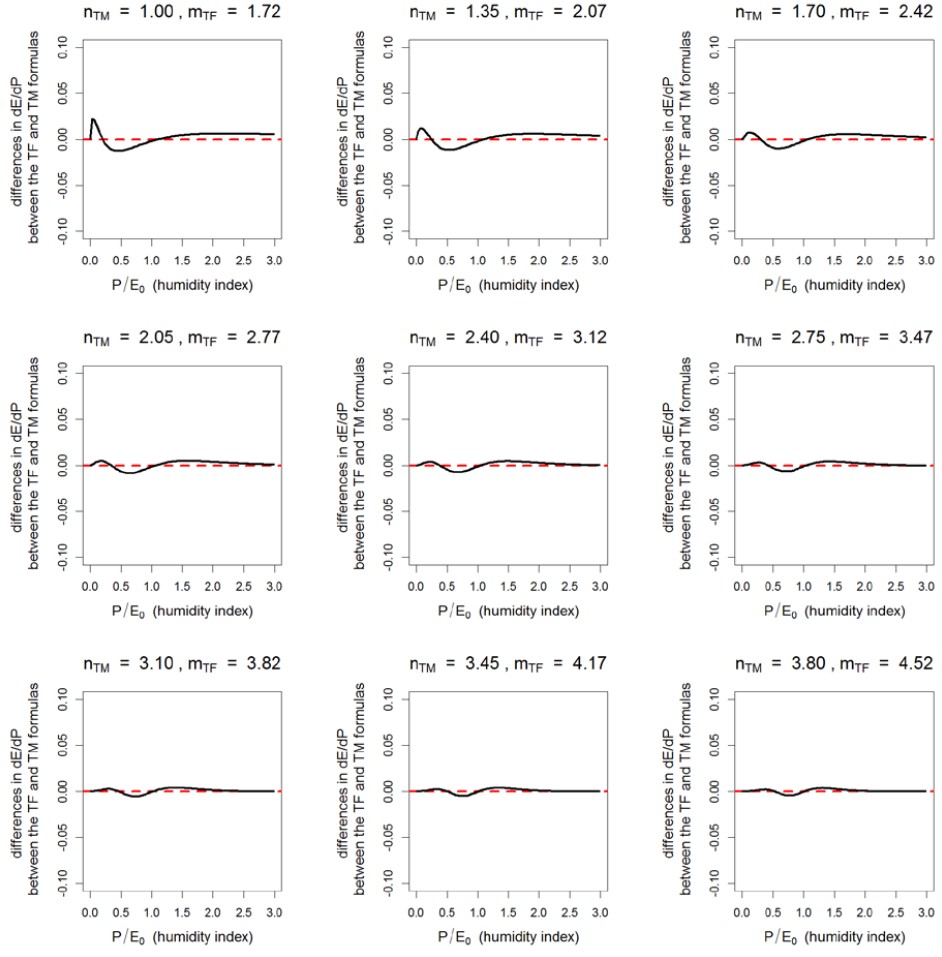

109

**Figure 2. Illustration of the similarity between the Turc-Mezentsev (TM) and the Tixeront-Fu (TF)**

**formulas for a range of values of *n* (the parameter of the TM formula) and *m* (the parameter of**

**the TF formula), using the Yang et al. (2008) relationship: *m* = *n* + 0.72 : difference in the partial**

**differentials $\frac{\partial E}{\partial P}$**

114



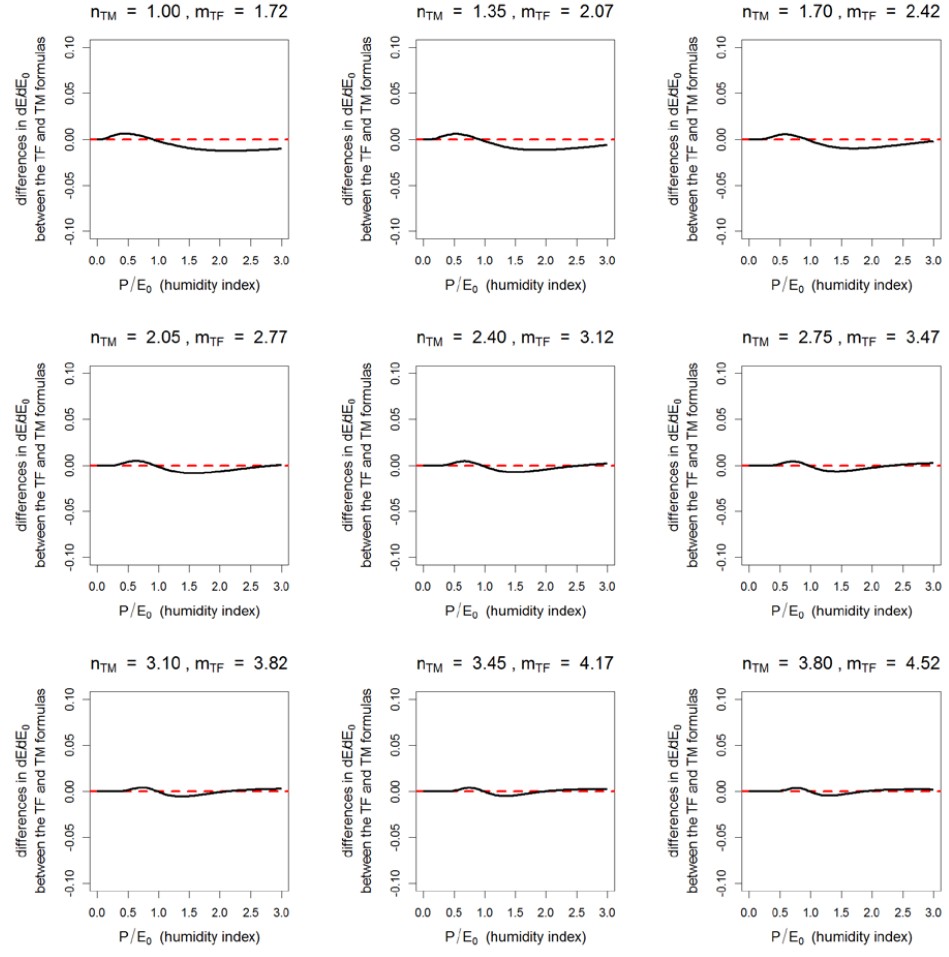

115

**Figure 3. Illustration of the similarity between the Turc-Mezentsev (TM) and the Tixeront-Fu (TF)**

**formulas for a range of values of *n* (the parameter of the TM formula) and *m* (the parameter of**

**the TF formula), using the Yang et al. (2008) relationship: *m* = *n* + 0.72 : difference in the partial**

**differentials $\frac{\partial E}{\partial E_0}$**

120

121



## 4. Interpretation of the TM and TF formulas

### 4.1 Hydrological interpretation

The TM and TF formulas share a set of hydrological properties that we summarize in Table 4 and Table 5, following the presentation proposed by Lebecherel et al. (2013).

**Table 4. Hydrological interpretation of the Turc-Mezentsev and Tixeront-Fu formulas, applied to streamflow ($P$ – precipitation, $Q$ – streamflow, $E_0$ – maximum evaporation, all in mm/year averaged over many years).**

| | Mathematical property | Hydrological interpretation |
|---|---|---|
| 1 | $Q < P$ | A catchment cannot produce more water than it receives from precipitation |
| 2 | $P - Q < E_0$ | A catchment cannot lose more water than it receives energy to evaporate water |
| 3 | $Q/P \to 1$ when $P \gg E_0$ | Water yield of very humid catchments tends towards 1 |
| 4 | $Q/P \to 0$ when $E_0 \gg P$ | Water yield of very arid catchments tends towards 0 |
| 5 | $\dfrac{\partial Q}{\partial P} \to 1$ when $P \gg E_0$ | On very humid catchments, all additional precipitation tends to be transformed into streamflow |
| 6 | $\dfrac{\partial Q}{\partial E_0} \to -1$ when $P \gg E_0$ | On very humid catchments, all additional energy tends to be subtracted from streamflow |
| 7 | $\dfrac{\partial Q}{\partial P} \to 0$ when $E_0 \gg P$ | On very arid catchments, streamflow is not sensitive to additional precipitation |
| 8 | $\dfrac{\partial Q}{\partial E_0} \to 0$ when $E_0 \gg P$ | On very arid catchments, streamflow is not sensitive to additional energy |

**Table 5. Hydrological interpretation of the Turc-Mezentsev and Tixeront-Fu formulas, applied to actual evaporation (P – precipitation, $E_0$ – maximum evaporation, E – actual evaporation, all in mm/year averaged over many years).**

| | Mathematical property | Hydrological interpretation |
|---|---|---|
| 1 | $E < P$ | A catchment cannot evaporate more water than it receives from precipitation |
| 2 | $E < E_0$ | A catchment cannot evaporate more water than it receives energy |
| 3 | $E \to P$ when $E_0 \gg P$ | Very arid catchments tend to use all incoming rainfall for evaporation |
| 4 | $E \to E_0$ when $P \gg E_0$ | Very humid catchments tend to use all incoming energy for evaporation |
| 5 | $\dfrac{\partial E}{\partial P} \to 0$ when $P \gg E_0$ | On very humid catchments, actual evaporation is not sensitive to additional precipitation |
| 6 | $\dfrac{\partial E}{\partial E_0} \to 1$ when $P \gg E_0$ | On very humid catchments, all the additional energy tends to be transformed into evaporation |
| 7 | $\dfrac{\partial E}{\partial P} \to 1$ when $E_0 \gg P$ | On very arid catchments, all the additional precipitation tends to be transformed into evaporation |
| 8 | $\dfrac{\partial E}{\partial E_0} \to 0$ when $E_0 \gg P$ | On very arid catchments, actual evaporation is not sensitive to additional energy |





### 4.2 Mathematical interpretation

The appendix summarizes the underlying mathematical reasoning presented by the authors of the TM and TF formulas and by Zhang et al. (2004) and Yang et al. (2008). What can be concluded from the analysis presented in the appendix is that both formulations are based on very similar but nonetheless slightly different hypotheses; Table 6 illustrates them after rewriting the partial differentials to make $E$ appear (for the TM formula see Yang et al., 2008, and Eq. 31 and Eq. 32 in appendix; for the TF formula see Fu, 1981, and Eq. 25 and Eq. 26 in the appendix):

- For the Turc-Mezentsev formula, Table 6 shows that $\frac{\partial E}{\partial P}$ and $\frac{\partial E}{\partial E_0}$ can only be written as functions of the $\frac{P}{E}$ and $\frac{E_0}{E}$ ratios;

- For the Tixeront-Fu formula, Table 6 shows that $\frac{\partial E}{\partial P}$ and $\frac{\partial E}{\partial E_0}$ can be written as functions of the $\frac{P}{E}$ and $\frac{E_0}{E}$ ratios (as for the TM formulation). But in addition, $\frac{\partial E}{\partial P}$ can be written a function of $\frac{E_0-E}{P}$ (i.e., the remaining energy divided by $P$) and $\frac{\partial E}{\partial E_0}$ can be written as a function of $\frac{P-E}{E_0}$ (the remaining water divided by $E_0$). In fact, Fu (1981) demonstrated in a rigorous mathematical way that the TF formulation was the only possible solution to this set of hypotheses (i.e., Eq. 22 in the appendix).

**Table 6. Comparison of the partial differentials of the Turc-Mezentsev and the Tixeront-Fu formula (P – precipitation, $E_0$ – maximum evaporation, E – actual evaporation, all in mm/year averaged over many years; $n$ is the free parameter of the Turc-Mezentsev formula [$n$ >0]; $m$ is the free parameter of the Tixeront-Fu formula [$m$ >1])**

| | Turc-Mezentsev formulation | Tixeront-Fu formulation | |
|---|---|---|---|
| $\frac{\partial E}{\partial P}$ | $\left(\frac{P}{E}\right)^{-1}\left[1-\left(\frac{E_0}{E}\right)^{-n}\right]$ | $1-\left[1+\left(\frac{P}{E}\right)^{-1}\left(\frac{E_0}{E}-1\right)\right]^{1-m}$ | $1-\left(1+\frac{E_0-E}{P}\right)^{1-m}$ |
| $\frac{\partial E}{\partial E_0}$ | $\left(\frac{E_0}{E}\right)^{-1}\left[1-\left(\frac{P}{E}\right)^{-n}\right]$ | $1-\left(1+\frac{\frac{P}{E}-1}{\frac{E_0}{E}}\right)^{1-m}$ | $1-\left(1+\frac{P-E}{E_0}\right)^{1-m}$ |
| | **Expression using $\frac{P}{E}$ and $\frac{E_0}{E}$ ratios** | | **Expression using $\frac{P-E}{E_0}$ and $\frac{E_0-E}{P}$ ratios** |

What can we conclude from this? Does this make the TF formula (slightly) more general and the TM formula (slightly) more restrictive? Perhaps, but from the user's point of view, both formulas are so close numerically (see Figure 1 and also compare the maps presented by de Lavenne and Andréassian, 2018) that any data-based distinction is impossible.






### 4.3    Mathematico-hydrological interpretation

We can suggest another interpretation of both equations, which we label "mathematico-
hydrological." For this, we need to define two simple functions, which we may tentatively call
"$D_{min}$ – minimum by default" and "$E_{max}$ – maximum by excess." Let x and y be strictly positive
quantities:

$$Dmin_n(x,y) = [x^{-n} + y^{-n}]^{\frac{-1}{n}} \qquad \text{Eq. 15}$$

$$Emax_m(x,y) = [x^{m} + y^{m}]^{\frac{1}{m}} \qquad \text{Eq. 16}$$


$Dmin_n$ gives *the minimum by default* because for all positive values of parameter *n* it returns
a value that is lower than the minimum of *x* and *y* and larger than 0. When *n* is large, $Dmin_n$
returns a value that is very close to the minimum of *x* and *y*. $Emax_m$ gives *the maximum by*
*excess* because for all positive values of parameter *m* it returns a value that is larger than the
maximum of *x* and *y*. When *m* is large, $Emax_m$ returns a value that is very close to the
maximum of *x* and *y*. Only for values of *m* greater than 1 is the value taken by $Emax_m$
smaller than the sum of *x* and *y*.
We can now hydrologically interpret the TM formula by saying that it states that catchment-
scale actual evaporation *E* is equal to the *minimum by default* of the forcing fluxes, $E_0$ and *P*.
Similarly, the interpretation of the TF formula is that *E* is equal to the sum of the forcing
fluxes, $E_0$ and *P*, minus their *maximum by excess*. A positive *E* requires *m* to be greater than
one.

## 5.    Conclusion

The Turc-Mezentsev and Tixeront-Fu formulas are two sound and numerically equivalent
representations of the long-term water balance at the catchment scale. This note
investigated the underlying assumptions of the two formulas and showed that the Tixeront-Fu
formula is slightly more general than the Turc-Mezentsev formula, because its partial
differences can be written both as a function of the $\frac{P}{E}$ and $\frac{E_0}{E}$ ratios and as a function of the
$\frac{E_0-E}{P}$ and $\frac{P-E}{E_0}$ ratios (the TM formula can only write its partial differences as a function of the $\frac{P}{E}$
and $\frac{E_0}{E}$ ratios). Apart from this difference, both formulas share the same hydrological
properties and we can see no reason to recommend one over the other as more
"hydrologically founded." This should not be considered disappointing: it simply means that
hydrologists should feel free to choose the formula they feel more comfortable with.




## 6.    Acknowledgements

The authors gratefully acknowledge the review provided by Dr Charles Perrin.

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






## 8. Appendix: Genealogy of the Turc-Mezentsev and the Tixeront-Fu formulations

### 8.1 Turc formula

Lucien Turc was a French soil scientist. He produced his formula while working on his PhD
thesis, defended in April 1953 (and published in 1954 in the *Annales Agronomiques*). Turc
used water balance data for a set of 254 catchments from all over the world, collected with
the help of Prof. Maurice Pardé, a well-known hydrologist of that time. He computed
catchment-scale long-term average actual evaporation ($E$) from estimates of long-term
average precipitation ($P$) and long-term average discharge ($Q$) by writing $E = P - Q$ (all
variables in mm/yr), and he used a polynomial relationship to compute $E_0$ from temperature.
After plotting his catchment data in the $E/E_0=f(P/E_0)$ nondimensional space, Turc looked for a
mathematical function running through the experimental points and respecting the two
following constraints:
• $\frac{E}{E_0} \sim \frac{P}{E_x}$ when $\frac{P}{E_0}$ is small
• $\frac{E}{E_0} \sim 1$ when $\frac{P}{E_0}$ is large
Turc (1954, p. 504) wrote that the simplest function respecting these two conditions would
be:
$y = \frac{x}{1+x}$,        with $y = \frac{E}{E_0}$ and $x = \frac{P}{E}$
and that the most general would be:

$$y = \frac{x}{(1+x^n)^{\frac{1}{n}}}, \text{ i.e., } \frac{E}{E_0} = \frac{\frac{P}{E_0}}{\left[1+\left(\frac{P}{E_0}\right)^n\right]^{\frac{1}{n}}} \text{ or } \frac{E}{P} = \frac{1}{\left[1+\left(\frac{P}{E_0}\right)^n\right]^{\frac{1}{n}}}$$      **Eq. 17**


in which $n$ is an exponent to estimate. Turc graphically looked for the most convenient value
for $n$ and concluded that the best fit was "with $n$=3, or maybe $n$=2" (Turc, 1954, p. 563). Since
the choice of $n$=2 allowed the simplest computations, he retained this value for further
developments.

### 8.2 Mezentsev formula

Varfolomeï Mezentsev was a Soviet geographer, working at the University of Omsk in
Siberia. He published his formula in 1955, and continued working on it throughout his life



(Mezentsev, 1955, 1982, 1993). Mezentsev started his analysis from a formula proposed by
Bagrov (1953) (Eq. 18):

$$\frac{dE}{dP} = 1 - \left(\frac{E}{E_0}\right)^n$$      **Eq. 18**

The Bagrov formula can be interpreted as follows: when $\frac{E}{E_0}$ is small, i.e., when water is the
limiting factor, an increase in precipitation $P$ is integrally transformed into an increase of
actual evaporation $E$. Conversely, when $\frac{E}{E_0}$ approaches 1 (i.e., when water does not limit
evaporation) none of the additional $P$ is transformed into $E$ because no more energy is
available for evaporation. Bagrov showed that this formula presents the interesting property
of integrating into the Oldekop (1911) water balance formula for $n$=2. For $n$=1, $n$=4/3 and
$n$=3/2, Bagrov found analytical solutions, but he could not find a generic solution for all
values of $n$.
Mezentsev (1955) reasoned that in order to find a generic solution, Bagrov's formula could
be rewritten as follows:

$$\frac{dE}{dP} = \left[1 - \left(\frac{E}{E_0}\right)^n\right]^{1+\frac{1}{n}}$$      **Eq. 19**

which keeps the same interpretation as Eq. 18.
Eq. 19 can be integrated analytically and yields Eq. 20:

$$\frac{E}{P} = \frac{1}{\left[1 + \left(\frac{P}{E_0}\right)^n\right]^{\frac{1}{n}}}$$      **Eq. 20**

which is identical to the general formulation proposed by Turc (i.e., Eq. 20, Eq. 17 and Eq. 2
are identical). Based on a set of 35 catchments of the Siberian plain, Mezentsev suggested
using the value of 2.3 for parameter $n$, which is also close to the value chosen by Turc.

**8.3     Tixeront formula**
Jean Tixeront (1901–1984), a graduate of Ecole Nationale des Ponts et Chaussées, was a
French hydrologist who spent most of his professional career in Tunisia. The most accessible
reference for his formula is a paper published in the proceedings of the General Assembly of
the IAHS in 1964 (Tixeront, 1964). The formula had been first published in 1958, in the note
accompanying a map of mean annual runoff in Tunisia (Berkaloff and Tixeront, 1958). There,
the authors give more explanation on their reasoning, stating that two desirable properties of
such a formula would be that (i) "when precipitation increases, runoff tends to equal
precipitation minus potential evapotranspiration" and (ii) "when precipitation tends towards



zero, the runoff to the precipitation ratio tends towards zero." They proposed Eq. 21 as the
"simplest formula satisfying these conditions":

$$Q = [P^m + E_0^m]^{\frac{1}{m}} - E_0 \qquad \text{Eq. 21}$$

Unfortunately, Tixeront never published the detailed computations that led him to the
formula.

## 8.4   Fu's system of differential equations

Bao-Pu Fu was a Chinese hydrologist working at the University of Nanjing. He published his
formula in 1981, and an English abstract of his computation is given in the appendix of the
paper by Zhang et al. (2004). It is interesting to note that Fu's paper (1981) starts with a well-
informed review of the formulas in the literature, where he cites the works of Bagrov (1953)
and Mezentsev (1955). Then he makes assumptions on a system of differential equations
that should be respected by an actual evaporation formula (eq. A1 in Zhang's paper):

$$\begin{cases} \dfrac{\partial E}{\partial P} = F(u) \\[2mm] \dfrac{\partial E}{\partial E_0} = G(v) \end{cases} \qquad \text{Eq. 22}$$

where $u$ and $v$ are given by

$$u = \frac{E_0 - E}{P} \ and \ v = \frac{P - E}{E_0} \qquad \text{Eq. 23}$$


The mathematical integration of the system given in Eq. 22 with the boundary conditions
given by lines 5, 6, 7 and 8 in Table 5 led to the following formula, which is equivalent (in
actual evaporation terms) to Tixeront's formula (i.e., Eq. 24 below and Eq. 4 are the same):

$$E = P + E_0 - [P^m + E_0^m]^{\frac{1}{m}} \qquad \text{Eq. 24}$$

Actually, from Eq. 10 and Eq. 4, it can easily be seen that:

$$\frac{\partial E}{\partial P} = 1 - P^{m-1}(P^m + E_0^m)^{\frac{1-m}{m}} = 1 - P^{m-1}(P + E_0 - E)^{1-m}$$

Therefore:

$$\frac{\partial E}{\partial P} = 1 - \left(1 + \frac{E_0 - E}{P}\right)^{1-m} \qquad \text{Eq. 25}$$

Similarly, from Eq. 12 and Eq. 4, it can easily be seen that:

$$\frac{\partial E}{\partial E_0} = 1 - E_0^{m-1}(P^m + E_0^m)^{\frac{1-m}{m}} = 1 - E_0^{m-1}(P + E_0 - E)^{1-m}$$

Therefore:





$$\frac{\partial E}{\partial E_0} = 1 - \left(1 + \frac{P - E}{E_0}\right)^{1-m}$$

Eq. 26

Hence, Eq. 25 and Eq. 26 show that the Tixeront function is indeed the solution of the Fu
system of differential equations in Eq. 22, with the following functions:

$$F(u) = 1 - (1 + u)^{1-m}, \quad G(v) = 1 - (1 + v)^{1-m}$$

Eq. 27


### 8.5 Yang et al.'s system of differential equations

Yang et al. (2008) were not only the first to compare the Turc-Mezentsev and the Tixeront-Fu
formulas, they also made a mathematical analysis of the Turc-Mezentsev formula, that we
reflect on now. They start to write down a system of differential equations that should be
respected by an actual evaporation formula (Eq. (14) in their 2008 paper):

$$\begin{cases} \dfrac{\partial E}{\partial P} = f(x, y) \\ \dfrac{\partial E}{\partial E_0} = g(x, y) \end{cases}$$

Eq. 28


where $x$ and $y$ are given by:

$$x = \frac{P}{E}, y = \frac{E_0}{E}$$

Eq. 29

The mathematical integration of the system given in Eq. 28 with the boundary conditions
given in lines 5, 6, 7 and 8 of Table 5 led to the following formula, which is equivalent to the
Turc-Mezentsev formula (i.e., Eq. 30 below and Eq. 2 are the same):

$$E = [P^{-n} + E_0^{-n}]^{\frac{-1}{n}}$$

Eq. 30

Actually, from Eq. 6 it is easily seen that:

$$\frac{\partial E}{\partial P} = P^{-n-1}(P^{-n} + E_0^{-n})^{\frac{-1}{n}-1} = \frac{(P^{-n} + E_0^{-n})^{\frac{-1}{n}}}{P} \frac{P^{-n}}{P^{-n} + E_0^{-n}}$$

Therefore, using Eq. 2 we have:

$$\frac{\partial E}{\partial P} = \frac{E}{P}\left(1 - \frac{E_0^{-n}}{E^{-n}}\right)$$

Eq. 31

Similarly, from Eq. 8 it is easy to see that:

$$\frac{\partial E}{\partial E_0} = E_0^{-n-1}(P^{-n} + E_0^{-n})^{\frac{-1}{n}-1} = \frac{(P^{-n} + E_0^{-n})^{\frac{-1}{n}}}{E_0} \frac{E_0^{-n}}{P^{-n} + E_0^{-n}}$$

Therefore, using Eq. 2 we have:



$$\frac{\partial E}{\partial E_0} = \frac{E}{E_0}\left(1 - \frac{P^{-n}}{E^{-n}}\right)$$

**Eq. 32**

Hence, Eq. 31 and Eq. 32 show that the Turc-Mezentsev function is indeed a solution of the
Yang et al. system of differential equations (Eq. 28) with the following functions:

$$f(x,y) = x^{-1}(1 - y^{-n}), \qquad g(x,y) = y^{-1}(1 - x^{-n})$$

**Eq. 33**


We wish to underline that the Turc-Mezentsev function is not the only solution of the Yang et
al. system of differential equations (Eq. 28). This system is also satisfied by the Tixeront-Fu
function. Indeed, $u$ and $v$ defined in Eq. 23 can also be expressed using the $x$ and $y$ ratios
defined in Eq. 29:

$$\frac{E_0 - E}{P} = \frac{E_0}{E}\frac{E}{P} - \frac{E}{P} = \frac{y-1}{x} \qquad , \qquad \frac{P - E}{E_0} = \frac{P}{E}\frac{E}{E_0} - \frac{E}{E_0} = \frac{x-1}{y}$$

Therefore, Eq. 25 and Eq. 26 show that Tixeront-Fu's formula satisfies the following
conditions:

$$\frac{\partial E}{\partial P} = 1 - \left(1 + \frac{y-1}{x}\right)^{1-m} \qquad \frac{\partial E}{\partial E_0} = 1 - \left(1 + \frac{x-1}{y}\right)^{1-m}$$

These formulas show that Tixeront-Fu's function is a solution of the Yang et al. system of
differential equations (Eq. 28) with the following functions:

$$f(x,y) = 1 - \left(1 + \frac{y-1}{x}\right)^{1-m} \qquad , \qquad g(x,y) = 1 - \left(1 + \frac{x-1}{y}\right)^{1-m}$$

**Eq. 34**

Thus, when Yang et al. (2008) wrote in their conclusion (p.8) that "this paper mathematically
derived the general solution to the mean annual water-energy balance equation, and proved
its uniqueness" this is obviously an error. It is interesting to look where in their demonstration
they "missed" the Tixeront-Fu formulation (which they knew perfectly). In their integration of
Eq. 28, these authors used the following computations. Assuming $P$ and $E_0$ are independent,
the differentiation of Eq. 28 gives the following formulas:

$$\frac{\partial^2 E}{\partial E_0 \partial P} = -\frac{x}{E}g\frac{\partial f}{\partial x} + \frac{1-yg}{E}\frac{\partial f}{\partial y}$$

$$\frac{\partial^2 E}{\partial P \partial E_0} = -\frac{y}{E}f\frac{\partial g}{\partial y} + \frac{1-xf}{E}\frac{\partial g}{\partial x}$$

A solution of Eq. 28 must satisfy the equation:

$$\frac{\partial^2 E}{\partial E_0 \partial P} = \frac{\partial^2 E}{\partial P \partial E_0}$$

Hence (Eq. (15) in the Yang et al. paper):



$$-xg\frac{\partial f}{\partial x} + (1 - yg)\frac{\partial f}{\partial y} = -yf\frac{\partial g}{\partial y} + (1 - xf)\frac{\partial g}{\partial x}$$

**Eq. 35**

Assume that functions $f$ and $g$ satisfy both Eq. (16a) and Eq. (16b) in the Yang et al. paper:

$$xg\frac{\partial f}{\partial x} = yf\frac{\partial g}{\partial y}$$

**Eq. 36**

$$(1 - yg)\frac{\partial f}{\partial y} = (1 - xf)\frac{\partial g}{\partial x}$$

**Eq. 37**

Then they obviously satisfy Eq. 35. However, the general solution of Eq. 35 does not
necessarily satisfy both Eq. 36 and Eq. 37. The computations given in Yang et al. (2008)
consist in solving these equations. They show that the functions given by Eq. 33 satisfy both
Eq. 36 and Eq. 37.
Straightforward computations show that the functions given by Eq. 34 do not satisfy Eq. 37,
although they satisfy Eq. 36. This is the reason why Yang et al. (2008) missed the solution
given by Tixeront-Fu's formula in their demonstration. For the functions $f$ and $g$ given by Eq.
34 we have:

$$xg\frac{\partial f}{\partial x} = (1 - m)\left(1 - \left(1 + \frac{x - 1}{y}\right)^{1-m}\right)\left(1 + \frac{y - 1}{x}\right)^{-m}\left(\frac{y - 1}{x}\right)$$

$$yf\frac{\partial g}{\partial y} = (1 - m)\left(1 - \left(1 + \frac{y - 1}{x}\right)^{1-m}\right)\left(1 + \frac{x - 1}{y}\right)^{-m}\left(\frac{x - 1}{y}\right)$$

Therefore:

$$xg\frac{\partial f}{\partial x} \neq yf\frac{\partial g}{\partial y}$$

so that Eq. 37 is not satisfied. On the other hand we have:

$$-xg\frac{\partial f}{\partial x} + (1 - yg)\frac{\partial f}{\partial y} = (m - 1)\left(1 + \frac{y - 1}{x}\right)^{1-m}\left(1 + \frac{x - 1}{y}\right)^{1-m}\frac{1}{x + y - 1}$$

$$-yf\frac{\partial g}{\partial y} + (1 - xf)\frac{\partial g}{\partial x} = (m - 1)\left(1 + \frac{y - 1}{x}\right)^{1-m}\left(1 + \frac{x - 1}{y}\right)^{1-m}\frac{1}{x + y - 1}$$


Therefore Eq. 36 is satisfied.