# Peer review of "Technical Note: On the confounding similarity of two water 2 balance formulas – Turc-Mezentsev vs Tixeront-Fu"

_Hydrology and Earth System Sciences, 2019_

## Referee Comment (RC1) · Anonymous Referee #1 · 5 Feb 2019

This technical note explores the similarity between two commonly-used Budyko-type water balance equations; namely the Turc-Mezentsev and the Tixeront-Fu versions. By several comparisons it is shown that Turc-Mezentsev and Tixeront-Fu are numerically near-identical. It is concluded that therefore both equations are equally good to use.

The technical note is straightforward and there is little to disagree with. However, in its present form, I find it difficult to see the novelty and relevance of this work, and I'd like to see them better clarified. Responding to the following points may help to do so:

- The main point that the Turc-Mezentsev and the Tixeront-Fu are near equivalent has been established previously by Yang et al. (2008). Why is it worth repeating this point?

[Figure]

What is really the novel addition of this work?

- What is the point of section 4.3: I read this section several times, but the description is not clear enough (for me) to understand what the value is of this paragraph (and I suspect other readers may suffer from the same problem as me).

Detailed suggestions

Line 1: I am unsure that "confounding" is really useful here. Would removing this word not make the title simpler, more accurate, and more objective? The same applies for every time the word "confounding" is used throughout the manuscript.

Line 13: "identified" seems redundant?

L36: why "maximum evaporation", rather than "potential evaporation"? The latter term seems more consistent with commonly used hydrological terminology.

L65-66: Explain why.

L88: Is this a result from this paper, or is this sourced from literature?

Table 4, property7: this statement is true for "absolute streamflow changes", not for "relative streamflow changes (i.e. streamflow elasticity)". Be explicit about this difference.

L138-140: explain in simple terms what is different.

Section 4.3: I don't understand the point of this section.

---

## Referee Comment (RC2) · Anonymous Referee #1 · 18 Feb 2019

I would like to thank the authors for their quick and satisfying response to my comments.

- My point that "Table 4, property7: this statement is true for "absolute streamflow changes", not for "relative streamflow changes (i.e. streamflow elasticity)" referred to the fact that a precipitation change in arid location will (on average) lead to a greater change relative change in runoff. For example, a 10% increase in P in an arid location could lead to a ∼20% increase in Q. Whereas in a very humid location, a 10% increase in P will lead to a ∼10% increase in Q. However, in hindsight I think this is not really relevant as you state the math next to the descriptive statement, so it should be clear that your statement refers to absolute runoff changes (which indeed will be smaller in

arid places).

- (While I am not a native English speaker) the current use of the word "confounding" does not seem to lead to clear (or correct) title. Because the two equations are near-identical, and this sometimes leads to confusions when they are used, one could say something like "the distinction between these two equations is confounded". However, saying "the equations are confounded" seems illogical to me. Anyway, I do not think this is a big deal, but I would encourage considering using a simple title that avoids any possible confusion.

---

## Referee Comment (RC3) · Maik Renner (Referee) · 5 Mar 2019

The manuscript by Andréassian and Sari explains the historical background of two well known formulations which describe the partitioning of water and energy balances under climatological average conditions. They also clarify the naming of these formulas and I believe that this note can help to achieve a more consistent usage of the two formulas in the literature.

The appendix on the genealogy of the two formulations is quite a treasure and I have a small concern that it might be overseen. I think that this appendix could be a section in the main text. Only the subsection on Yang's system is a bit long, but indeed very

interesting.

The paper is very well written and thereby provides a clear and easy to follow discussion of the hydrological interpretation and the mathematical derivation. Hence this paper will be a valuable source for hydrologists which need orientation in the vast literature on that topic.

Minor remark: Figures: the limits of the y-axis could be decreased to better see the differences. In the moment there is too much unused space.

---

## Author Comment (AC2) · 6 Mar 2019

Thank you very much for the time you spent reviewing our paper. We will introduce the historical part in the main body of the paper.

---

## Referee Comment (RC4) · Laurène Bouaziz (Referee) · 7 Mar 2019

The authors provide a comprehensive and well-written comparison of two independently derived water balance formulas: Turc-Mezentsev versus Tixeront-Fu. The authors show that the two formulas are numerically equivalent (also in their partial differentials), and even though the Tixeront-Fu formula can be characterized as slightly more general, hydrologists can feel free to choose either one of them. An interesting analogy is made between the mathematical characteristics of the shape of the formulas and their hydrological meaning. Additionally, the Appendix provides an overview of the history and derivation of the formulas. I enjoyed reading this comprehensive comparison of the two water balance formulas with a clear final message and I therefore recommend the publication of this manuscript after only a few minor corrections.

Comments:

- Line 24: Apostrophe s is missing in: "Turc's work"

- Line 86: 'than' instead of 'that'?

- Line 97: It is mentioned that both formulas are equivalent except for very low values of the humidity index and I wonder if there is an explanation to this observation.

- Section 4.3 (line 163-180): This section makes an interesting mathematical analysis of the hydrological formulas, but it would make it easier for the reader to explicitly refer to Eq. 2 and Eq. 4 to explain the analogy with Eq. 15 and Eq. 16.

- Line 255: I believe a typo was introduced in this formula and that the authors meant $E/E0 \sim P/Eo$ instead of $P/Ex$

- Line 259: here also I think a typo was introduced and that the formula should read $x = P/E0$ instead of $x = P/E$

---

## Author Comment (AC3) · 26 Mar 2019

Thank you for your careful review, and especially for helping detect the two typos in the formulas.

When you ask whether there is an explanation to the fact that both formulas only differ for very low humidity indices, we must recognize that we could not think of any mathematical explanation.

Concerning section 4.3 : Thank you for the encouraging words. We are not sure that we will keep this section as it is, because reviewer 1 found it extremely difficult to

understand. We had thought that interpreting the two formulas as an approximation of the classical Min and Max functions would help the reader "visualize" what the formulas were doing. . . but it seems that it remained too abstract? Perhaps could we move it to the appendices...

---

## Referee Comment (RC5) · Laurène Bouaziz (Referee) · 30 Mar 2019

Thank you for your reply. In my opinion, Section 4.3 will not remain too abstract if an explicit reference to Equations 2 and 4 is made to make the analogy with Equations 15 and 16. Indeed, it would be good to keep this section in the Appendix.

---

## Author Comment (AC4) · 17 Apr 2019

Dear Mrs Bouaziz,

Thank you for the advice. We will keep this section in the Appendix.

---

## Author Comment (AC5) · 17 Apr 2019

Thank you once again for your comments. As already stated, we will remove the adjective confounding from the main text. As far as the title is concerned, we propose to replace "confounding" by "puzzling", which conveys the same meaning but without the risk of misunderstanding.

---

## Author Response (AR1)

We provide here a detailed answer to the questions raised by the reviewers (same answers were posted in the public discussion).

**REVIEWER 1**

1. The main point that the Turc-Mezentsev and the Tixeront-Fu are near equivalent has been established previously by Yang et al. (2008). Why is it worth repeating this point? What is really the novel addition of this work?

We completely agree that Yang et al. (2008) established the equivalence, and we do give them proper credit for it in our note. However, we do consider that their paper was not clear on a few points, and this is why we saw a need for a « clarifying » technical note. We find the Yang et al. paper unclear/incomplete on the following points:

- Equivalence between the two equations: Yang et al. write that the TM and TF equations are « approximately equivalent », we find the expression much too weak and this is why we wished to use the much stronger « confounding » ;
- Literature review: Yang et al. make no reference to the original work of Turc (1954) and Tixeront (1964). They likely were not aware of it ;
- Uniqueness of solution: Yang et al. wrote in their conclusion (p.8) that "this paper mathematically derived the general solution to the mean annual water-energy balance equation, and proved its uniqueness". This is obviously wrong (and to tell the truth this is extremely surprising because Yang et al. are comparing the TM and TF formulas, they know perfectly that the solution is not unique) and this is why we added table 6 to show that the TF formula respects both hypotheses.

Last, in our note we tried to treat as much as possible the two forms of the formulas in parallel (streamflow & actual evaporation) to provide a reference for those who wish to use one or the other.

2. What is the point of section 4.3: I read this section several times, but the description is not clear enough (for me) to understand what the value is of this paragraph (and I suspect other readers may suffer from the same problem as me).

Section 4.3 was an attempt to explain with a lot of words and little formulas what the TM and TF represented. This was not easy and we know that the result is not perfect. If you did not understand it, it very likely means that we failed to explain clearly what we had on our minds. We will remove this part from the main text, put it in appendix with a note showing how the two functions relate to the TF and TM formulas..

Detailed suggestions

3. Line 1: I am unsure that "confounding" is really useful here. Would removing this word not make the title simpler, more accurate, and more objective? The same applies for every time the word "confounding" is used throughout the manuscript.

We added "confounding" precisely because we thought that Yang et al. had not been affirmative enough when stating that both formulas were "approximately equivalent". But we take your point on the fact that this word is perhaps useful in the title, but not anymore is the rest of the paper: we did remove it elsewhere, and replace it by "puzzling" in the title

4. Line 13: "identified" seems redundant?

Yes indeed, removing it does simplify the sentence.

5. L36: why "maximum evaporation", rather than "potential evaporation"? The latter term seems more consistent with commonly used hydrological terminology.

The hydrologists usually use only "potential evaporation" while the agronomists distinguish theoretical potential evaporation/potential evaporation/actual evapotranspiration/maximal actual evapotranspiration/potential (grass) evapotranspiration, etc. You are right that potential evaporation is more common in hydrology. Because the TM and TF formulas are considered as "Budyko-type" formulas, we wanted to utilize Budyko's formulation, i.e. maximum evaporation to avoid any debate with our colleagues agronomists.

6. L65-66: Explain why.

We could rewrite L 65-66 as follows:

"In our interpretation of the TM and TF formulas, we will also use their partial derivatives, which we present in Table 2 and Table 3 (they are sometimes used to predict the hydrological impact of climatic change)."

7. L88: Is this a result from this paper, or is this sourced from literature?

It is in fact in Yang et al. paper (which as cited a few lines above). We will add a reference.

8. Table 4, property7: this statement is true for "absolute streamflow changes", not for "relative streamflow changes (i.e. streamflow elasticity)". Be explicit about this difference.

We are not sure to understand this remark, because we would define the relative elasticity as the linear relationship between (Qn/Qmean -1) and (Pn/Pmean -1), with n an index for the year. Could you be more explicit?

9. L138-140: explain in simple terms what is different.

The detailed mathematical explanation comes a few lines later (LL 144-151) so for this sentence we could simply complement the sentence:

*What can be concluded from the analysis presented in the appendix is that both formulations are based on very similar but nonetheless slightly different hypotheses ;*

Into

*What can be concluded from the analysis presented in the appendix is that both formulations are based on very similar but nonetheless slightly different hypotheses, which set the dependency of the partial differences of streamflow to the partial differences of climatic variables ;*

10. Section 4.3: I don't understand the point of this section.

We tried to explain the behavior of the generalized harmonic mean with plain language, in a less mathematical way, but if you did not understand, this probably mean that it did not help to make think clearer, so we will put this short section in appendix

**REVIEWER 2 (Maik Renner)**

The manuscript by Andréassian and Sari explains the historical background of two well known formulations which describe the partitioning of water and energy balances under climatological average conditions. They also clarify the naming of these formulas and I believe that this note can help to achieve a more consistent usage of the two formulas in the literature. The appendix on the genealogy of the two formulations is quite a treasure and I have a small concern that it might be overseen. I think that this appendix could be a section in the main text. Only the subsection on Yang's system is a bit long, but indeed very interesting. The paper is very well written and thereby provides a clear and easy to follow discussion of the hydrological interpretation and the mathematical derivation. Hence this paper will be a valuable source for hydrologists which need orientation in the vast literature on that topic. Minor remark: Figures: the limits of the y-axis could be decreased to better see the differences. In the moment there is too much unused space.

We hesitated to introduce the historical part in the main text, but we did not find a way to do it that would not turn the paper too complex to read. We left it in appendix but added a sentence to encourage readers to go and read this part.

**REVIEWER 3 (Laurène Bouaziz)**

1. The authors provide a comprehensive and well-written comparison of two independently derived water balance formulas: Turc-Mezentsev versus Tixeront-Fu. The authors show that the two formulas are numerically equivalent (also in their partial differentials), and even though the Tixeront-Fu formula can be characterized as slightly more general, hydrologists can feel free to choose either one of them. An interesting analogy is made between the mathematical characteristics of the shape of the formulas and their hydrological meaning. Additionally, the Appendix provides an overview of the history and derivation of the formulas. I enjoyed reading this comprehensive comparison of the two water balance formulas with a clear final message and I therefore recommend the publication of this manuscript after only a few minor corrections.

**Comments:**
2. Line 24: Apostrophe s is missing in: "Turc's work" : done
3. Line 86: 'than' instead of 'that'? done
4. Line 97: It is mentioned that both formulas are equivalent except for very low values of the humidity index and I wonder if there is an explanation to this observation. We could not think of any mathematical explanation (and because these hyper-arid catchments are anyway extremely difficult to model, we stopped looking for it)

5. Section 4.3 (line 163-180): This section makes an interesting mathematical analysis of the hydrological formulas, but it would make it easier for the reader to explicitly refer to Eq. 2 and Eq. 4 to explain the analogy with Eq. 15 and Eq. 16. Thank you, however we are not sure that we will keep this section, reviewer 1 found it extremely difficult to understand. We found that interpreting the two formulas as an approximation of the classical Min and Max functions would help the reader "visualize" what the formula was doing… but it seems that it remained to abstract?

6. - Line 255: I believe a typo was introduced in this formula and that the authors meant $E/E_0 \sim P/E_0$ instead of $P/E_x$ - done, thank you

7. - Line 259: here also I think a typo was introduced and that the formula should read $x = P/E_0$ instead of $x = P/E$ - done, thank you

**Technical Note: On the  puzzling similarity of two water balance formulas – Turc-Mezentsev *vs* Tixeront-Fu**

Vazken Andréassian[1] & Tewfik Sari[2]

[1] Irstea, HYCAR Research Unit, Antony, France

[2] ITAP, Univ Montpellier, Irstea, Montpellier SupAgro, Montpellier, France

**Abstract**

This Technical Note documents and analyzes the  puzzling similarity of two widely used water balance formulas: Turc-Mezentsev and Tixeront-Fu. It details their history, their hydrological and mathematical properties, and discusses the mathematical reasoning behind their slight differences. Apart from the difference  in their partial differential expressions, both formulas share the same hydrological properties and it seems impossible to recommend one over the other as more "hydrologically founded": hydrologists should feel free to choose the one they feel more comfortable with.

**Keywords**

Water balance formulas, Turc-Mezentsev formula, Tixeront-Fu formula, Budyko hypothesis

**1. Introduction**

The Turc-Mezentsev (Mezentsev, 1955;Turc, 1954) and Tixeront-Fu (Fu, 1981;Tixeront, 1964) formulas were introduced to model long-term water balance at the catchment scale. Both formulas are almost equivalent numerically (but differ nonetheless). Surprisingly, comparisons are rare: Tixeront knew the work of Turc (1954) , which he cites, but it seems that he did not realize that Turc's formulation was numerically equivalent to the one he proposed. Similarly, Fu knew the work of Mezentsev (1955)  because he precisely starts his 1981 paper discussing it, but it seems that he did not realize that the formulation he obtained was so close numerically.

As far as we know, Yang et al. (2008) were the first to compare the Turc-Mezentsev and the Tixeront-Fu formulas and to conclude that both formulas were "approximately equivalent." In

this note we further elaborate the  similarity between the two formulas and contribute complementary explanations on their underlying hypotheses.

**2. Presentation of the Turc-Mezentsev (TM) and the Tixeront-Fu (TF) formulas**

The TM and TF formulas use as inputs long-term average precipitation $P$ [mm/yr] and long-term average maximum evaporation $E_0$ [mm/yr]. They produce as outputs either long-term average specific discharge $Q$ [mm/yr] or long-term average actual evaporation $E$ [mm/yr]. There are two formulations (one giving $Q$ as a function of $P$ and $E_0$ and one giving $E$ as a function of the same variables), equivalent under the assumption that the catchment is conservative (i.e., that it does not "leak" towards deep aquifers) so that $E$ and $Q$ can be linked through the equation $E = P - Q$. Maximum evaporation is understood in the sense of Budyko (1963 /1948/) as the water equivalent of the energy available to evaporation. In what follows, the $E_0/P$ ratio is called the aridity ratio, its inverse (i.e., the $P/E_0$ ratio) is called the humidity ratio. The formulas are presented in Table 1. Because none of the original papers introducing them are in English, we also  document their origins in the appendix, in order to provide interested readers with a more detailed description of the origine of each formula.

**Table 1. Turc-Mezentsev (TM) and Tixeront-Fu (TF) water–energy balance formulations ($P$ – precipitation, $Q$ – streamflow, $E_0$ – maximum evaporation, $E$ – actual evaporation, all in mm/year averaged over many years). $n$ is the free parameter of the Turc-Mezentsev formula [$n$ >0]; $m$ is the free parameter of the Tixeront-Fu formula [$m$ >1].**

| Reference | Streamflow formulation | Actual evaporation formulation | Parameter |
|---|---|---|---|
| Turc (1954), Mezentsev (1955) | $Q = P - [P^{-n} + E_0^{-n}]^{\frac{-1}{n}}$

Eq. 1 | $E = [P^{-n} + E_0^{-n}]^{\frac{-1}{n}}$

Eq. 2 | $n > 0$ |
| Tixeront (1964), Fu (1981) | $Q = [P^m + E_0^m]^{\frac{1}{m}} - E_0$

Eq. 3 | $E = P + E_0 - [P^m + E_0^m]^{\frac{1}{m}}$

Eq. 4 | $m > 1$ |

We need to clarify here that the TM and TF formulas can be found in the hydrologic literature under different names. The naming convention we adopted is explained as follows: Eq. 1 and Eq. 2 are named "Turc-Mezentsev" (TM) because Turc (1954) and Mezentsev

[revised manuscript text omitted]
 Ms Laurène Bouaziz, Dr Maik Renner, Dr Charles Perrin and an anonymous reviewer, which all contributed to clarify the manuscript.

used water balance data for a set of 254 catchments from all over the world, collected with
the help of Prof. Maurice Pardé, a well-known hydrologist of that time. He computed
catchment-scale long-term average actual evaporation ($E$) from estimates of long-term
average precipitation ($P$) and long-term average discharge ($Q$) by writing $E = P - Q$ (all
variables in mm/yr), and he used a polynomial relationship to compute $E_0$ from temperature.
After plotting his catchment data in the $E/E_0 = f(P/E_0)$ nondimensional space, Turc looked for a
mathematical function running through the experimental points and respecting the two
following constraints:

•      $\frac{E}{E_0} \sim \frac{P}{E_{\cancel{x}0}}$ when $\frac{P}{E_0}$ is small

•      $\frac{E}{E_0} \sim 1$ when $\frac{P}{E_0}$ is large

Turc (1954, p. 504) wrote that the simplest function respecting these two conditions would
be:

$y = \frac{x}{1+x}$,      with $y = \frac{E}{E_0}$ and $x = \frac{P}{\cancel{E}E_0}$

and that the most general would be:

$$y = \frac{x}{(1+x^n)^{\frac{1}{n}}}, \text{ i.e., } \frac{E}{E_0} = \frac{\frac{P}{E_0}}{\left[1+\left(\frac{P}{E_0}\right)^n\right]^{\frac{1}{n}}} \text{ or } \frac{E}{P} = \frac{1}{\left[1+\left(\frac{P}{E_0}\right)^n\right]^{\frac{1}{n}}}$$      **Eq. 15**

in which $n$ is an exponent to estimate. Turc graphically looked for the most convenient value
for $n$ and concluded that the best fit was "with $n=3$, or maybe $n=2$" (Turc, 1954, p. 563). Since
the choice of $n=2$ allowed the simplest computations, he retained this value for further
developments.

**8.2 Origin of the Mezentsev formula**

Varfolomeï Mezentsev was a Soviet geographer, working at the University of Omsk in Siberia. He published his formula in 1955, and continued working on it throughout his life (Mezentsev, 1955, 1982, 1993). Mezentsev started his analysis from a formula proposed by Bagrov (1953) (Eq. 16Eq. 18):

$$\frac{dE}{dP} = 1 - \left(\frac{E}{E_0}\right)^n \qquad \text{Eq. 16}\text{18}$$

The Bagrov formula can be interpreted as follows: when $\frac{E}{E_0}$ is small, i.e., when water is the limiting factor, an increase in precipitation $P$ is integrally transformed into an increase of actual evaporation $E$. Conversely, when $\frac{E}{E_0}$ approaches 1 (i.e., when water does not limit evaporation) none of the additional $P$ is transformed into $E$ because no more energy is available for evaporation. Bagrov showed that this formula presents the interesting property of integrating into the Oldekop (1911) water balance formula for $n$=2. For $n$=1, $n$=4/3 and $n$=3/2, Bagrov found analytical solutions, but he could not find a generic solution for all values of $n$.

Mezentsev (1955) reasoned that in order to find a generic solution, Bagrov's formula could be rewritten as follows:

$$\frac{dE}{dP} = \left[1 - \left(\frac{E}{E_0}\right)^n\right]^{1+\frac{1}{n}} \qquad \text{Eq. 17}\text{19}$$

which keeps the same interpretation as Eq. 16Eq. 18.

Eq. 17Eq. 19 can be integrated analytically and yields Eq. 18Eq. 20:

$$\frac{E}{P} = \frac{1}{\left[1 + \left(\frac{P}{E_0}\right)^n\right]^{\frac{1}{n}}} \qquad \text{Eq. 18}\text{20}$$

which is identical to the general formulation proposed by Turc (i.e., Eq. 18Eq. 20, Eq. 15Eq. 17 and Eq. 2Eq. 2 are identical). Based on a set of 35 catchments of the Siberian plain, Mezentsev suggested using the value of 2.3 for parameter $n$, which is also close to the value chosen by Turc.

**8.3 Origin of the Tixeront formula**

Jean Tixeront (1901–1984), a graduate of Ecole Nationale des Ponts et Chaussées, was a French hydrologist who spent most of his professional career in Tunisia. The most accessible reference for his formula is a paper published in the proceedings of the General Assembly of the IAHS in 1964 (Tixeront, 1964). The formula had been first published in 1958, in the note accompanying a map of mean annual runoff in Tunisia (Berkaloff and Tixeront, 1958). There, the authors give more explanation on their reasoning, stating that two desirable properties of such a formula would be that (i) "when precipitation increases, runoff tends to equal precipitation minus potential evapotranspiration" and (ii) "when precipitation tends towards zero, the runoff to the precipitation ratio tends towards zero." They proposed Eq. 19Eq. 21 as the "simplest formula satisfying these conditions":

$$Q = [P^{\mathrm{m}} + E_0^{\mathrm{m}}]^{\frac{1}{\mathrm{m}}} - E_0 \qquad \textbf{Eq. 1921}$$

Unfortunately, Tixeront never published the detailed computations that led him to the formula.

**8.4 On Fu's system of differential equations**

Bao-Pu Fu was a Chinese hydrologist working at the University of Nanjing. He published his formula in 1981, and an English abstract of his computation is given in the appendix of the paper by Zhang et al. (2004). It is interesting to note that Fu's paper (1981) starts with a well-informed review of the formulas in the literature, where he cites the works of Bagrov (1953) and Mezentsev (1955). Then he makes assumptions on a system of differential equations that should be respected by an actual evaporation formula (eq. A1 in Zhang's paper):

$$\begin{cases} \dfrac{\partial E}{\partial P} = F(u) \\ \dfrac{\partial E}{\partial E_0} = G(v) \end{cases} \qquad \textbf{Eq. 2022}$$

where $u$ and $v$ are given by

$$u = \frac{E_0 - E}{P} \;\; and \;\; v = \frac{P - E}{E_0} \qquad \textbf{Eq. 2123}$$

The mathematical integration of the system given in Eq. 20Eq. 22 with the boundary conditions given by lines 5, 6, 7 and 8 in Table 5Table 5 led to the following formula, which is equivalent (in actual evaporation terms) to Tixeront's formula (i.e., Eq. 22Eq. 24 below and Eq. 4Eq. 4 are the same):

$$E = P + E_0 - [P^{\mathrm{m}} + E_0^{\mathrm{m}}]^{\frac{1}{\mathrm{m}}} \qquad \textbf{Eq. 2224}$$

Actually, from Eq. 10Eq. 10 and Eq. 4Eq. 4, it can easily be seen that:

[revised manuscript text omitted]

Therefore Eq. 34Eq. 36 is satisfied.

## 8.6    Another interpretation of the Turc-Mezentsev and Tixeront-Fu formulas

We present here another interpretation of both equations, which is partly mathematical and
partly hydrological. For this, we define two simple functions, which we tentatively call "$Dmin$ –
minimum by default" and "$Emax$ – maximum by excess." Let $x$ and $y$ be strictly positive
quantities:

$$Dmin_n(x,y) = [x^{-n} + y^{-n}]^{\frac{-1}{n}}$$ **Eq. 36**

$$Emax_m(x,y) = [x^m + y^m]^{\frac{1}{m}}$$ **Eq. 37**

Obviously, $Dmin_n$ reminds Eq. 2 (the Turc-Mezentsev formulation) and $Emax_m$ reminds Eq.
3 (the Tixeront-Fu formulation). These two functions have interesting mathematical
properties which we can try to interpret also hydrologically:
$Dmin_n$ gives *the minimum by default* because for all positive values of parameter *n* it returns
a value that is lower than the minimum of *x* and *y* and larger than 0. When *n* is large, $Dmin_n$
returns a value that is very close to the minimum of *x* and *y*. $Emax_m$ gives *the maximum by*
*excess* because for all positive values of parameter *m* it returns a value that is larger than the
maximum of *x* and *y*. When *m* is large, $Emax_m$ returns a value that is very close to the
maximum of *x* and *y*. Only for values of *m* greater than 1 is the value taken by $Emax_m$
smaller than the sum of *x* and *y*.
We can now hydrologically interpret the TM formula by saying that it states that catchment-
scale actual evaporation *E* is equal to the *minimum by default* of the forcing fluxes, $E_0$ and *P*.
Similarly, the interpretation of the TF formula is that *E* is equal to the sum of the forcing
fluxes, $E_0$ and *P,* minus their *maximum by excess*. A positive *E* requires *m* to be greater than
one.